# Cold Response of the Mediterranean Fruit Fly (*Ceratitis capitata*) on a Lab Diet

**DOI:** 10.3390/insects10020048

**Published:** 2019-02-03

**Authors:** Farhan J.M. Al-Behadili, Vineeta Bilgi, Junxi Li, Penghao Wang, Miyuki Taniguchi, Manjree Agarwal, Yonglin Ren, Wei Xu

**Affiliations:** 1School of Veterinary and Life Sciences, Murdoch University, Murdoch, WA 6150, Australia; f.al-behadili@murdoch.edu.au (F.J.M.A.-B.); vineetabilgi@gmail.com (V.B.); J.Li@murdoch.edu.au (J.L.); P.Wang@murdoch.edu.au (P.W.); M.Taniguchi@murdoch.edu.au (M.T.); M.Agarwal@murdoch.edu.au (M.A.); 2College of Agriculture, Misan University, Misan 62001, Iraq

**Keywords:** Mediterranean fruit fly, *Ceratitis capitata*, cold treatment, postharvest treatment

## Abstract

Cold treatment at 0.0 °C with different exposure durations (0–12 days) was applied to the Mediterranean fruit fly *Ceratitis capitata* (Wiedemann) fed on a lab diet. The examined developmental stages were early eggs (<6 h), late eggs (>42 h), first instar, second instar and third instar larvae. Pupation, adult emergence and sex ratios of survived flies were investigated to study the *C. capitata* responses to this low temperature treatment. Our results showed that exposure time at low temperature has a clear effect on pupation and adult emergence. Based on pupation ratios, the first and third instar are the most cold tolerant stages, with LT99 = 7.3 for both of them. Cold tolerance at both stages are very close and no significant differences were detected. There were no significant differences on *C. capitata* sex ratios among different stages after treatment. This study improves our understanding of *C. capitata* responses to cold treatment, which may assist in the improvement of the current treatment strategies to control this destructive horticulture pest species.

## 1. Introduction

The Mediterranean fruit fly, *Ceratitis capitata* (Wiedemann) (Diptera: Tephritidae) is one of the most destructive and invasive insect pests for horticulture biosecurity, global trade and world-wide phytosanitary [1]. *C. capitata* originated from sub-Saharan Africa and then spread throughout the Mediterranean region, Europe, the Middle East, Western Australia and both South and Central America [2].

*C. capitata* has been recorded feeding on over 300 fruit, vegetable and nut plant species. The main hosts include citrus, stone fruits, pome fruits, peppers, tomatoes and figs [3]. Plant hosts also include avocadoes, apricots, persimmons, strawberries, grapes, bananas, bitter melons, carambolas, coffees, guavas, peppers, papayas and blueberries. After mating, one female adult *C. capitata* can lay as many as 800 eggs in her lifetime [4]. Since a *C. capitata* attack happens when the majority of the production cost have already been expended, they cause huge horticultural industry losses. Therefore, *C. capitata* are regarded as one of the most destructive horticulture pest species. Furthermore, in recent years, the growing international trade of plant products increased the risk of introducing fruit flies across countries. Strict government policies were quickly made to mitigate these risks and minimize the damages. For example, pre-harvest actions including spraying, monitoring and inspections [5], with postharvest treatments such as fumigation, irradiation, heat or cold are required to control fly species [6]. Nevertheless, the current global trade of farm fresh products is suffering from the damages by various fruit fly species, prompting the need to develop more effective ways of fruit fly control.

The available quarantine treatment technologies mainly consist of chemical (e.g., fumigation) and non-chemical treatments (e.g., cold, heat and irradiation). In recent years, with the discontinuation of several chemical products (e.g., Fenthion and Dimethoate), it has become even more urgent to rely on non-chemical postharvest control technologies to control fruit flies [7]. Cold treatment is becoming an increasingly popular postharvest treatment due to the absence of chemical residues, mitigations or mortality of the pest population, as well as increasing the strength of the fruits, and prolonging storage time [8,9,10]. It can also be applied to fruit at multiple stages, for example, after packing and ‘in transit’ during lengthy transport by sea, as well as co-treatment with other postharvest treatments such as irradiation [11].

Optimal cold treatment conditions rely on the commodities, and most previous studies on *C. capitata* cold treatment reflect this, performing on the flies while they are within the fruits [8,9,10,12,13,14,15,16]. On the other hand, because fruits vary in sizes, nutrients, compositions, and phytochemical profile, there were large differences in efficacy of heat transfer and fly development using similar cold treatment regime. For example, a previous study on the *C. capitata* cold tolerance in date and mandarin at 1.11 °C showed that *C capitata* is more sensitive to cold treatment when in date fruits than in mandarins [16]. The identification of the most cold tolerant development stage is crucial to determine the thoroughness of the treatment; however, there were also discrepancies on which *C. capitata* developmental stage was the most cold tolerant in previous reports. For example, Grout et al. [14] concluded that the most cold tolerant stage was the 2nd instar based on a commodity group research report from South Africa. Hallman et al. showed that the 3rd instar is the most cold tolerant stage [17]. We therefore find it compelling to conduct a research on cold tolerance of *C. capitata* reared on a lab diet. The result will provide a fundamental baseline for comparison with data obtained from those conducted within fruits, and will be essential to establish cold response model, considering fruit sizes, compositions, nutrients and other variables. The overall objective of this study was to evaluate and understand *C. capitata* responses to low temperature. Based on these understandings, our long-term goal is to optimize current postharvest treatments and develop more environmentally friendly, cost-effective and efficient treatments for controlling *C. capitata*.

## 2. Materials and Methods

### 2.1. Insect Culture

*C. capitata* used in this study was originally established in 2015 from a laboratory colony maintained at the Department of Primary Industries and Regional Development’s (DPIRD) in Western Australia, which has been periodically supplemented with the introduction of wild flies. Mature females lay eggs through the mesh (cloth sidewalls of the cages), which were collected and transported to the artificial rearing medium which consists of 1 kg ground dehydrated carrot, 300 g torula yeast, 4.5 L hot tap water, 36 mL HCL, 30 g nipagin and 500 mL boiling water [18]. After 13–16 days, pupae were collected and transferred into the adult breeding cages. The emerged adult flies were reared on the yeast hydrolysate, crystalline sugar and water. Rearing conditions were 26.0 ± 1.0 °C, 60–65% room humidity, and darkness light cycle of 16:8 h [13]. Early eggs (<6 h), late eggs (>42 h), 1st instar, 2nd instar and 3rd instar larvae were used in this study.

### 2.2. Cold Treatment Rooms

The cold room is a prefabricated unit with walls and ceilings of 100 mm expanded polystyrene. Joints and base of rooms are sealed with silicon sealant under aluminum covering extrusions. The floor is concrete; the door is 1500 mm wide × 1900 mm high × 100 mm thick with expanded polystyrene for insulation. The room dimensions are 4.33 m wide × 3.77 m length × 2.00 m high [11].

The refrigeration for the cold room is supplied by 2 × Patton (Model CCH 250, Patton, Auckland, New Zealand) air cooled Condensing Unit with R22 refrigerant + 2 × Patton BL 38 Induced Draught Evaporator with refrigeration capacity of 5090 Watts at 0 °C. The temperature of the room is controlled through a surface-mounted electronic thermostat (DIXEL, Roma, Italy) having a temperature range from −50 to +110 °C with a probe installed in the return air path. Up to 4× defrost cycles can occur per 24 h if required. Two fans in the room (300 mm 5 blade propeller types) circulate air across the evaporator at an air flow rate of approximately 960 L/s measured at various points in the room.

To maintain consistent temperature and to avoid temperature changes when opening the door of cold rooms, tested flies were placed in a chamber (1.2 m × 0.8 m × 0.6 m = 0.576 m^3^) located within the cold rooms. The temperatures inside the chamber were recorded every 30 min by Applent Multi-channel Temperature recorder (AT4508-128, Made-in-China.com, Nanjing-Jiangsu, China). The sensors were set inside the chamber in different levels. All sensors were calibrated in ice water (0 °C) before and after the treatment.

### 2.3. Cold Treatment Tests

The carrot medium was taken from a fridge approximately 24 h before usage and was placed in sterile 90 mm plastic petri dishes at room temperature. The petri dishes were covered with lids and set aside on a lab bench until infestation with insect of various stages. Approximately, 55 g of carrot medium was added per petri dish with a clean spoon, making sure that the medium was not aggregated or in contact with the lid. This ensured that larvae, if hatched, were able to move freely through the medium and were not stuck to the lid. The lids from the control petri dishes were removed when larvae were first observed during incubation; the same number of days were then applied for treatment petri dishes when removing the lids.

Freshly laid *C. capitata* eggs were collected over a period of 1 h prior to the trial and allowed to settle in a clean glass beaker containing 100 mL double distilled water. To count the number of eggs (in this study 100), droplets of water with suspending eggs were placed on a sterile petri dish using a transfer pipet. Eggs inside each droplet were counted using a manual cell counter, carefully picked up using a transfer pipet and placed onto the carrot medium. Once the petri dishes were ready with the eggs, they were placed in a tray and transported to the cold rooms, where they were carefully placed in stacks of six replicate per exposure time inside the acrylic box.

100 eggs or larvae were collected, counted and transferred onto one petri dish with the carrot diet. Six replicates were prepared to be exposed from 0 to 12 days at 0.0 °C (6 petri dishes ×100 = 600 egg or larvae each day). After treatment, the 6 petri dishes (replicates) containing eggs or larvae were retrieved at regular intervals of 24 h; then, each petri dish was placed in a 750 mL clear disposable container containing a layer of sand and kept in an incubation chamber at 26.0 ± 1.0 °C; 60–65% RH. Pupation and adult emerging over approximately 4 weeks were recorded. Untreated controls were kept in the same conditions (at 26.0 ± 1.0 °C; 60–65% RH) for pupae counting and adult emerging. In this study, the total number of each *C. capitata* stage experimented was 7800 (7200 for treatments + 600 for control); the final number of egg and larvae tested in the whole experiment was 39,000.

A piece of mesh cloth was put on top of the container, and the lid was then affixed. The lids were prepared to have six holes ensuring air circulation when placed in the incubator. Control petri dishes were checked regularly, and on day 8, when instars were visible, the lid was removed and the remaining configuration was left as described. The lids of the petri dishes in treatment groups were removed after being placed in the incubator as follows: after 7–8 days for early eggs trial; after 5–6 days for late eggs; after 3–4 days for first instar larvae; and after 1–2 days for second instar larvae. Third instar larvae had no lids from the start. During each trial, the temperatures were recorded at 30 min intervals in each cold room, at the carrot diet, and in the air of the chamber. The sand was renewed three times over six weeks to collect pupae. Pupae, emerged adults and the sex ratios were recorded and analyzed.

To study the effects of cold treatment on eggs, mortality was recorded by comparing a total number of pupae or adults produced, and eggs that were incubated at 26.0 °C were used as a control. This comparison provided mortality, and eggs were claimed “live” if pupae or adults were produced after treatment. This procedure included sieving through sand containing pupae followed by counting the total number of pupae. Sieving was carried out using a metal mesh tray (1.6 mm) that allowed sand particles to pass through but retained pupae. Sieving was carried out as soon as pupae were first seen, with this process repeated three times until there were no more pupae found in the sand. Pupae were carefully placed on a clean surface, and any large sand particles that remained were separated using a glass slide. Pupae were then counted and placed in a sterile petri dish which sat on a laboratory bench at room temperature. Additional observations on the number of emerged flies were considered. This provided information on the viability of pupae and whether they were able to develop into adults.

To study if the cold treatment affected the sex development of the emerged adults, we compared the sex ratios of the treated eggs and larvae. We collected all the survival adults after treatments from five developmental stages and calculated the percentage of female.

### 2.4. Statistical Analysis

The mortality rate of the insect under cold treatment was statistically estimated following the Median lethal time method (LT). The 90% and 99% mortality (LT90 and LT99) were estimated by using the selected models. We evaluated four different models separately for pupae and adults on every stage, including eggs, 1st instar, 2nd instar and 3rd instar. The best fitting model was selected for estimating the LT90 and LT99. The LT estimated under generalized linear model with both a probit and a logit link function on cold treatment days, which are the mostly used dose–response model where in our scenario days of treatment was considered as dose. The model can be written as:η = β_0 + β_1 x(1)
where η is the response or proportion mortality, x is the dose, β0 is the intercept and β1 is the coefficient of the dose. The four evaluated models include (1) probit model on log transformed treatment days; (2) logit model on log transformed treatment days; (3) probit model on treatment days without log transformation; and (4) direct logit model on treatment days. The best model was selected based on three different criteria, including exploratory analysis, Bayesian information criterion (BIC) value and regression residues. The best fitting model following these criteria for each insect stage was finally selected to estimate LT90 and LT99. This demonstrates that we are using the best fitting model for LT estimation. A 95% confidence interval was also reported. R statistical environment (version 3.3.2, R Foundation for Statistical Computing, Vienna, Austria, http://www.R-project.org/) with base library was used to estimate the LT and confidence interval, a ggplot2 package was used for generating plots. An ANOVA single factor test was used to compare sex ratios of adult insects emerging from different stages after cold treatment.

## 3. Results

In this project, a total of ~39,000 *C. capitata* eggs or larvae were used for cold treatment. Exactly 7800 eggs/larvae were used for each stage (early eggs, late eggs, 1st instar, 2nd instar and 3rd instar larvae). The duration of treatment ranged from 0 to 12 days at 0.0 °C to reach 100% mortality (Table 1). During the cold treatment, probes were used to monitor the temperature in the cold room, and the results showed that the temperature was stable at 0.0 ± 0.2 °C, from the start to the end of the experiment.

The effects of cold treatment on *C. capitata* are shown in Table 1 and Figure 1. Pupation and adult emergence ratios were used to calculate the mortality rates. By using pupation, if egg/larvae could not develop to pupae after treatment, this was defined as “death”. Similarly, by using adult emergence, if egg/larvae could not develop to an adult after treatment, this was defined as “death”. Our results showed that the five developmental stages (early egg, late egg, 1st instar, 2nd instar and 3rd instar) differed in their cold tolerance at 0.0 °C. By using the recovered pupation (Table 1 and Appendix A), 1st instar larvae was the most cold-tolerate stage, as it took nine days at 0.0 °C in the lab diet to reach zero pupae (100% mortality). The 3rd instar is the second most cold-tolerate stage, where took seven days at 0.0 °C to reach 0 pupae. Early eggs, late eggs and 2nd instar need six days to reach zero pupae after treatment. Interestingly, using the emerged adults to study the mortality achieved slightly different results (Table 1). For example, early eggs and 2nd instar all needed six days to achieve zero adults (100% mortality) from 600 eggs/larvae. Late eggs and 3rd instar both needed five days to achieve zero emerged adults, thus suggesting late eggs have a greater susceptibility to cold than early eggs. The 1st instar larvae stage is the most cold-tolerate stage, as it took the most days (9 days) to reach 100% mortality.

The pupation or adult emergence ratios from the non-treated (control) eggs/larvae varied (Appendix A). The results for larvae stages (1st, 2nd and 3rd instar) are very similar (89–99% pupation and 77–89% adult emergence). The early eggs in the control showed 69.5% pupation and 66.6% adult emergence, while the late eggs showed only 46.8% pupation and 32.3% adult emergence. The control eggs (including both early and late eggs) showed much lower pupation/adult emergence ratios than the control larvae (1st, 2nd and 3rd instar), which may be due to the low hatching ratios from eggs to larvae (Table 1).

We then further compared the pupation ratio and adult emergence ratio at each treated fly developmental stage (Figure 1). Our results showed that most mortality occurred before the pupation stage, as most insects that survived pupation would also emerge to adults. However, slight post-pupation mortality rates were observed in early egg (Figure 1A) and late egg (Figure 1B). While the post-pupation mortality in 1st instar and 2nd were higher than eggs. In the 3rd instar treatment, post-pupation mortality rates were the highest (Figure 1E), suggesting a considerable proportion of recovered pupae from this stage did not successfully develop to adults.

As mentioned, four different models were selected separately for pupations and emerged adults on every stage, and the best fitting model was selected for estimating the LT90 and LT99 (Appendix A). The four models include: log on days with probit, log on days with logit, no-log on days with probit, and no-log on days with logit (Appendix A). The results showed that no-log models are better than log models in this study (Appendix A). When using pupation as the end point for mortality analysis, no-log on days with probit model was selected for early egg, late egg and 3rd instar while no-log on days with logit was selected for 1st and 2nd instar larvae. When using emerged adults as the end point for mortality analysis, no-log on days with probit model was selected for late egg while no-log on days with logit was selected for the rest stages. We modelled the duration of cold treatment to induce 90 and 99% mortality at five immature stages of *C. capitata* fed on a lab diet (Table 2). Using the recovered pupation as the end point for mortality modeling, the results showed that the 3rd instar and 1st are the most cold tolerant stages, with LT99 = 7.3 days for both of instars. Interestingly, based on the emerged adult ratios, LT99 was 5.7, 5.7, 6.4, 5.4, and 6.1 days for early eggs, late eggs, 1st instar, 2nd instar and 3rd instar, respectively. The 1st instar (LT99 = 6.4) is superior to 3rd instar (LT99 = 6.1). However, the statistical analysis by using SPSS software showed that there are no significant differences between 1st and 3rd instars in either pupation or emerged adults as end points.

In terms of the sex ratio studies, female adults (%) developed from treated early eggs, late eggs, 1st, 2nd and 3rd instar were 51.7, 48.8, 52.2, 51.8 and 48.4, respectively. The ANOVA single factor test (Appendix A) showed that there are no significant differences between each stage. Overall, cold treatment at 0.0 °C on *C. capitata* does not affect sex ratio.

## 4. Discussion

Cold treatment is a common method for eradicating fruit flies in fresh fruits and other products. It has been studied, analysed and incorporated into quarantine regulations [6,8,10,11] and oversea countries [7,12,13,14].

In this study, we chose to study *C. capitata* cold response from the lab diet but not fruit, due to the variation in sizes, materials, nutrients and chemicals, resulting in significant differences in fly physiology and development. Secondly, some fruit may not be favorable hosts for fruit flies and so might be associated with the production of weak flies, resulting in the differences in the mortality rates. Furthermore, there are different infestation methods for various fruit. For example, by using the natural infestation, how many eggs were laid into a single fruit is hard to clarify. By using artificial infestation (e.g., injecting eggs or larvae into fruits), methods may affect fruit quality, fruit fly development and may even cause bacterial or fungi contamination. All of these can result in very different results in cold treatment experiments. Therefore, here we chose to use lab diets, as the high homogeneity of the diet in terms of ingredients, quantity, size and dimensions of experimental units make the results more reliable and reducing of experimental error. This way, we could be clear about how many eggs/larvae were used in the analysis and avoid the fruit/infestation issues. Cold treatment analysis of *C. capitata* on a lab diet have previously been performed [15], but not in a comparative way on all five different stages as we did here.

To evaluate cold treatment effects on *C. capitata*, the first question is how to define mortality. Eggs or larvae may survive from the cold treatment, but fail in pupation or adult emergence due to unknown effects. On the other hand, some of them succeed in adult emergence but fail in sex development and reproduction, so they are not biologically “alive” flies. Therefore, here we used the recovered pupae or adults as standards to help define our mortality here. If one egg or larva fails in developing to pupae or adult after treatment, it is a “dead” fly.

Our bioassays (Figure 1 and Table 1) showed that the 1st instar is the most cold tolerate stage. However, modelling analysis LT99 results showed the 1st and 3rd instar are equal stages in tolerance of cold treatment based on pupation. Based on the emerged adults, LT99 results showed that there is no significant differences between 1st and 3rd instars, in spite of 1st instar LT99 = 6.4 being superior to 3rd instar, because the range of the confidence interval range is 0.5 day. All these results suggest both 1st instar and 3rd instar are among the most cold tolerant stages, on which more attention should be paid in our postharvest treatment. The previous studies are not consistent concerning the most cold tolerant stage of *C. capitata*. There are several major reviews and seven annexes of International Standards for Phytosanitary Measures (ISPM) that deal specifically with the cold tolerance of this species. Grout et al. [12] concluded that the most cold tolerant stage was the 2nd instar based on a commodity group research report from South Africa. Hallman et al. showed that the 3rd instar is the most cold tolerant stage [15]. A study compared tolerance of eggs, a mixture of 1st and 2nd instars, and mostly 3rd instars to 1.5 ± 0.5 °C in oranges and found both larval groups to be very similar, with the younger instars showing a very slight advantage in survival [19]. Another study found that the 2nd instar was the most tolerant to 1.0 ± 0.2 °C in two cultivars of lemon [8]. The third study found that the 2nd instar was more tolerant than the 3rd in five types of citrus fruits at 2 and 3 °C [17]. There are plentiful differences in the experiment set up, including, but not limited to fruits, infestation methods, temperatures, fruit sizes, materials and nutrient. Therefore, it is not surprising that the detected most cold tolerate stage is different. It is also likely that different population of fruit flies vary in relative tolerance of the different states to cold treatment. It was reported previously that geographically isolated populations of Medfly differ in reproductive patterns, survival, developmental rates and intrinsic rates of increase [20], suggesting their different tolerance to low temperatures.

In this study, we compared early eggs (< 6 h) and late eggs (>42 h) in the cold treatment, because the embryo development stage may affect their biology, physiology and cold tolerance. For the fruit picked up and transferred to the cold storage, it is likely to collect some early eggs, which were laid just before the fruit picking, and late eggs, which have been laid for a period of time. The egg is a fast-developing stage, so the age difference can be a significant issue that has been ignored in previous research. Here, our results showed that early eggs are a bit more tolerant than late eggs.

The exposure durations required to achieve complete mortality of *C. capitate* immature stages have been reported in various fruits. For example, at 2 °C, 18 days were required to control *C. capitate* in mandarins and oranges, compared to 16 days in lemons; whereas at 3 °C, 20 days were required in mandarins and oranges, and 18 days were required in lemons [21]. At 0.5–1.5 °C, 16 days were needed to control *C. capitate* in mandarins [22]. Our study showed that nine days on a carrot diet was enough to achieve 100% mortality of *C. capitate* immature stages. The duration is much shorter, because we used an artificial diet, which is very different from real fruit in many ways. For example, the cold transfer in fruit can take more time than in artificial diet. Another reason is that we used 0 °C, lower than the temperatures used in many previous studies. Since there is a close association between duration of exposure, temperature and survival in most insects, the lower temperature leads to the faster injury and mortality.

We also compared the sex ratios of survived adults after treatments. Our result (Appendix A) showed that there is no significant difference in the sex ratios at different stages. While a number of studies investigate industry-relevant applications of cold treatment of Tephritid fruit flies [5,6,7,12] in various fruits at various stages, a fundamental understanding of the mechanisms underlying why and how cold can kill the flies is currently lacking. Without this knowledge, it is difficult for us to improve and optimize our current cold treatment strategies and develop more efficient treatment methods.

## 5. Conclusions

In this study, cold treatment at 0.0 °C with different exposure durations (0–12 days) was applied to the Mediterranean fruit fly *C. capitata* (Wiedemann) fed on a lab diet. The examined developmental stages were early eggs (<6 h), late eggs (>42 h), 1st instar, 2nd instar and 3rd instar larvae. Our results showed that both 3rd instar and 1st instar are among the most cold tolerant stages of *C. capitata*. There were no significant differences between early eggs and late eggs in the tolerance of cold treatment. We studied stage-specific tolerance because in infested fruit experiments, it is probably unlikely that all fruit flies are in the same developmental stage. Therefore, any treatment should be efficient enough to kill the most tolerant stage. There were no significant differences on *C. capitata* sex ratios among different stages after treatment. In future studies on cold treatments of fruit flies, we recommend further investigation of the interaction between survival times and test temperatures. Different temperatures may induce different responses at cellular and molecular levels. This study will improve our understanding of *C. capitata* responses to cold treatment and assist in the optimization of current treatment strategies to control this horticulturally destructive pest species.

## Figures and Tables

**Figure 1 insects-10-00048-f001:**
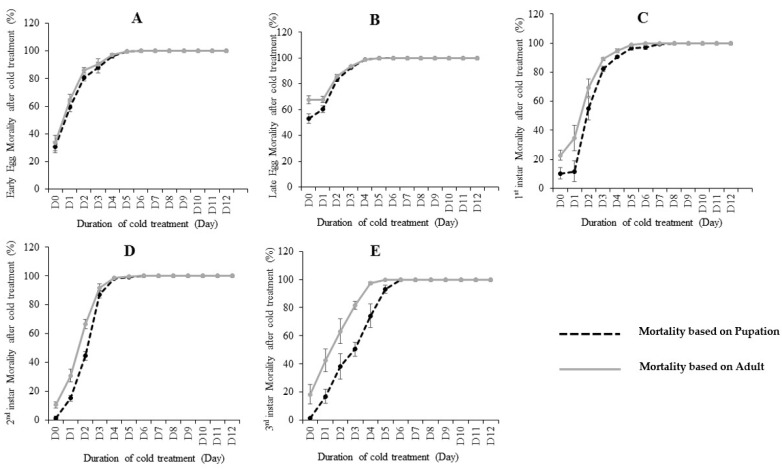
The mortality (%) calculated based on pupation and emerged adults at different immature stages including: early eggs (**A**), late eggs (**B**), 1st instar (**C**), 2nd instar (**D**) and 3rd instar (**E**) larvae of *C. capitata* fed on lab diet when subjected to various days of cold treatment at 0.0 °C. Error bar means standard error.

**Table 1 insects-10-00048-t001:** Pupation and emerged adult ratios of early eggs (< 6 h), late eggs (>42 h), 1st instar, 2nd instar and 3rd instar larvae of *C. capitata* fed on lab diet when subjected to various days of cold treatment at 0.0 °C. *n* = 6. SE: standard error.

Treatment		Early Egg		Late Egg		1st Instar		2nd Instar		3rd Instar	
Days	Insect Numbers	Pupation (%)	Emerged Adult (%) (Mean ± SE)	Pupation (%) (Mean ± SE)	Emerged Adult (%) (Mean ± SE)	Pupation (%) (Mean ± SE)	Emerged Adult (%) (Mean ± SE)	Pupation (%) (Mean ± SE)	Emerged Adult (%) (Mean ± SE)	Pupation (%) (Mean ± SE)	Emerged Adult (%) (Mean ± SE)
		(Mean ± SE)									
0	600	69.5 (4.2)	66.6 (5.7)	46.8 (3.6)	32.3 (3.0)	89.6 (4.1)	77.3 (3.5)	98.5 (1.1)	89.3 (2.0)	98.3 (0.6)	81.6 (6.9)
1	600	40.5 (3.6)	35.5 (3.8)	39.6 (2.4)	32.1(2.3)	79.1 (6.9)	65.5 (8.7)	83.5 (1.9)	69.0 (4.6)	81.6 (4.9)	57.5 (8.1)
2	600	19.5 (2.2)	14.0 (1.9)	16.8 (1.4)	14.1 (1.5)	40.5 (7.5)	30.6 (5.8)	54.5 (2.9)	33.3 (3.0)	60.8 (9.1)	36.8 (8.1)
3	600	12.0 (3.7)	9.7 (4.0)	7.3 (0.5)	6.1 (0.3)	16.0 (1.7)	11.0 (1.0)	12.5 (2.6)	8.1 (2.5)	48.6 (5.0)	18.5 (2.9)
4	600	3.8 (1.6)	2.6 (1.0)	1.3 (0.4)	1.3 (0.4)	8.3 (0.8)	5.3 (1.2)	1.6 (0.6)	1.3 (0.5)	25.3 (8.3)	2.6 (0.84)
5	600	0.3 (0.21)	0.2 (0.2)	0.2 (0.2)	0	3.1 (1.3)	1.6 (0.6)	0.8 (0.3)	0.3 (0.2)	6.6 (2.9)	0
6	600	0	0	0	0	2.8 (1.3)	0.3 (0.3)	0	0	0.3 (0.2)	0
7	600	0	0	0	0	0.7 (0.7)	0.3 (0.3)	0	0	0	0
8	600	0	0	0	0	0.3 (0.3)	0.2 (0.2)	0	0	0	0
9	600	0	0	0	0	0	0	0	0	0	0
10	600	0	0	0	0	0	0	0	0	0	0
11	600	0	0	0	0	0	0	0	0	0	0
12	600	0	0	0	0	0	0	0	0	0	0

**Table 2 insects-10-00048-t002:** Cold treatment duration to induce 90 and 99% mortality based on pupation and emerged adults of five *C. capitata* developmental stages fed on lab diet. The models were selected based on the result in Appendix A.

Mortality (LT)%	Developmental Stage	Pupae Recovery as End Point			Adults Recovery as End Point		
		Treatment (day)	95% Confidence Limits		Treatment (day)	95% Confidence Limits	
			Lower	Upper		Lower	Upper
LT90%	Early eggs	3.9	3.8	4	3.5	3.4	3.6
	Late eggs	3.5	3.4	3.7	3.3	3.2	3.5
	1st instar	5.2	5.1	5.3	4.2	4.1	4.4
	2nd instar	4.1	4.1	4.2	3.8	3.7	3.9
	3rd instar	5.7	5.6	5.8	4.2	4.1	4.3
LT99%	Early eggs	5.7	5.5	6	5.7	5.4	6
	Late eggs	5.5	5.3	5.8	5.7	5.4	6
	1st instar	7.3	7.1	7.5	6.4	6.1	6.6
	2nd instar	5.4	5.3	5.6	5.4	5.2	5.6
	3rd instar	7.3	7.2	7.5	6.1	5.9	6.3

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
