# Peer review of "Cold Response of the Mediterranean Fruit Fly (Ceratitis capitata) on a Lab Diet"

_insects, 2019, doi:10.3390/insects10020048_

Round 1

Reviewer 1 Report

General Comments:

This paper examines the stage specific cold tolerance of Mediterranean fruit fly. This is an important pest and low temperature is common tool to control this quarantine pest. Much of the data in this paper confirms data that is already available in the literature. The authors do good job of summarizing this literature. Their work is slightly different than the other papers in that they use artificial diets rather than fruit. I am not familiar with rearing this insects does Gazit et al 2014 use artificial diets?

The authors concentrate in the discussion covering if there are differences between the life stages in their work and in the literature. I would be interested in that they compare the durations needed to control these insects. I believe their work with artificial diets gives similar estimates as the work done with whole fruits.

The authors make the following statements.

Ln 246 “By using loss of emerged adults as end point, our bioassay showed that 1st instar is the most cold tolerant (LT99=6.4) and 3rd instar is the second most cold tolerant stage (LT99=6.14). However, by using loss of pupae as end point, the modelling analysis results showed 3rd instar is the most cold tolerant stage (LT99=7.36) and 1st instar is the second most cold tolerant stage (LT99=7.33).”

This is not true. Although, 6.4 is not the same number as 6.14, these numbers are not significantly different given the confidence intervals. The whole paper needs to be reviewed to state what is significantly different.

In this same vein, I would give all data as single digits, eg 6.1 d vs 6.14 d, given the range of the confidence interval range is 0.5 day.  

I believe Table 2 and Fig. 1 is the same data set. Present one of these. My suggestion would be to keep the graph. Given there is no significant differences in sex, I would delete Figure 2. Give some means and SEM and the statistics in the text.

Specific Comments:

I don’t think Table 1 is needed, this could be said in one or two sentences in the text.

Probit statistics often give; slope, intercept, n and chisquared, eg Hallman et al 2011. I suggest the authors do the same in Table S1, Table 3.

Author Response

I attached our response as word document

Reviewer 2 Report

In the paper, different Medfly stages, reared on an artificial diet, were tested for cold treatment at daily time exposures. Several papers were published on this argument, the originality of the present one is that Authors assessed mortality of individuals reared on artificial diet, not influenced by the characteristics of different fruits. The paper is well written, the experiments are correctly conducted, material and methods and results are clearly exposed. I have no much more to say as general comments: some additional references in the Introduction are required concerning the Medfly biology, not only limited to Australian cases. In the final part of the discussion, genomic researches are introduced, with various new technologies today available, but it is not clear what is the connection in the context of the presented research. In the References section, the used format does not meet in many cases the requested style (for example do not use et al.).

Minor remarks:

In Abstract: do not use abbreviations like E.E. or L.E., 1st …

Line 26: C. capitata in italic

Line 28: reference [2] is old, look a more recent title for distribution.

From line 30 to line 38 and lines 42-45: add some references

Lines 74-75: shortly describe the composition of the rearing medium.

Line 75: Tanaka… assign a reference number and add in the Reference section

Line 98: Carrot media maybe is “carrot medium”?

Line 115: the sentence in brackets is an important point, take out of the brackets and make more explicit that individuals recovered at different times belong to different samples.

Line 179: “pupils” not clear

Lines 243-244: this sentence is a comment, move to discussion

Table 3 and Figure 2 captions: C. capitata in italics

Lines 254-256: this explanation is misleading: in reptile sex is determined by temperatures, in fruit flies there are sexual chromosomes.

Lines 255-258: this part must be moved to Methods

Lines 262-263: report, here or as supplementary material, the results of ANOVA (F, df, sign.)

Figure 2: letter B is not reported in the graphic.

Line 358: Ceratitis capitata in italics and shortened

Line 361: Ceratitis capitata in C. capitata

Line 366: correct C. capitate in C. capitata

Reference [1]: M., W.I. and E.-H.M., M.  correct

Check separate scientific names in references n. 7,8,16

S1 Table caption: C. capitate correct in C. capitata

S1 Table: not clear: Method: “pupil or adult” only on the first row.

S1 Table: “Which its blow a line…: not clear

Author Response

I attached our response as word document

Insects EISSN 2075-4450 Published by MDPI AG, Basel, Switzerland RSS E-Mail Table of Contents Alert
Back to Top